# PriME-PGx: La Princesa University Hospital Multidisciplinary Initiative for the Implementation of Pharmacogenetics

**DOI:** 10.3390/jcm10173772

**Published:** 2021-08-24

**Authors:** Pablo Zubiaur, Gina Mejía-Abril, Marcos Navares-Gómez, Gonzalo Villapalos-García, Paula Soria-Chacartegui, Miriam Saiz-Rodríguez, Dolores Ochoa, Francisco Abad-Santos

**Affiliations:** 1Clinical Pharmacology Department, La Princesa University Hospital, Instituto Teófilo Hernando, Instituto de Investigación Sanitaria La Princesa (IP), Universidad Autónoma de Madrid (UAM), 28029 Madrid, Spain; ginapaola.mejia@scren.es (G.M.-A.); marcos.navares@salud.madrid.org (M.N.-G.); g.villapalos@salud.madrid.org (G.V.-G.); paulasch98@gmail.com (P.S.-C.); mdolores.ochoa@salud.madrid.org (D.O.); 2UICEC Hospital Universitario de La Princesa, Plataforma SCReN (Spanish Clinical Research Network), Instituto de Investigación Sanitaria La Princesa (IP), 28006 Madrid, Spain; 3Research Unit, Fundación Burgos por la Investigación de la Salud (FBIS), Hospital Universitario de Burgos, 09006 Burgos, Spain; miriam.saiz@salud.madrid.org; 4Centro de Investigación Biomédica en Red de Enfermedades Hepáticas y Digestivas (CIBERehd), Instituto de Salud Carlos III, 28200 Madrid, Spain

**Keywords:** clinical pharmacogenetics, implementation, precision medicine, personalized treatment

## Abstract

The implementation of clinical pharmacogenetics in daily practice is limited for various reasons. Today, however, it is a discipline in full expansion. Accordingly, in the recent times, several initiatives promoted its implementation, mainly in the United States but also in Europe. In this document, the genotyping results since the establishment of our Pharmacogenetics Unit in 2006 are described, as well as the historical implementation process that was carried out since then. Finally, this progress justified the constitution of La Princesa University Hospital Multidisciplinary Initiative for the Implementation of Pharmacogenetics (PriME-PGx), promoted by the Clinical Pharmacology Department of Hospital Universitario de La Princesa (Madrid, Spain). Here, we present the initiative along with the two first ongoing projects: the PROFILE project, which promotes modernization of pharmacogenetic reporting (i.e., from classic gene-drug pair reporting to complete pharmacogenetic reporting or the creation of pharmacogenetic profiles specific to the Hospital’s departments) and the GENOTRIAL project, which promotes the communication of relevant pharmacogenetic findings to any healthy volunteer participating in any bioequivalence clinical trial at the Clinical Trials Unit of Hospital Universitario de La Princesa (UECHUP).

## 1. Introduction

Pharmacogenetics is the medical discipline born in the 1950s that studies the role of genetic variation affecting drug response or adverse reactions to drugs [1]. Implemented in the clinical practice, this discipline helps to bring a personalized treatment to each patient. Consequently, ineffective or potentially toxic treatments are avoided or optimized. Unfortunately, during the past decades, the implementation of pharmacogenetics in clinical practice was limited for various reasons. Firstly, the lack of consistency in clinical recommendations and of the usefulness of pharmacogenetics. Secondly, the budgetary constraints that impeded routine large-scale genotyping of patients. Thirdly, the difficulty of interpreting pharmacogenetic information or the lack of specialists in the field. Fourthly, the lack of training of prescribing physicians and pharmacists, which made them hesitant to trust this discipline. However, by 2020, the situation was very different: there were several clinical pharmacogenetic guidelines from different scientific societies or consortia, some of them with very high levels of evidence. These include the Consortium for the Implementation of Clinical Pharmacogenetics (CPIC), the Dutch Pharmacogenetics Working Group (DPWG), among others, who base their clinical recommendations on a comprehensive compilation of scientific evidence. Moreover, the cost of genetic testing was significantly reduced; for instance, the complete sequencing of the human genome was worth approximately 100 million dollars in 2001, whereas nowadays it costs approximately 600 dollars [2]. Nowadays, specialists in pharmacogenetics capable of interpreting the clinical guidelines have been trained and are able to give sound therapeutic recommendations. As a consequence, physicians and pharmacists are nowadays much more willing to apply pharmacogenetics in the management of patients. Furthermore, the pharmacoeconomic repercussion of pharmacogenetics implementation in the clinical practice has been studied widely; in conclusion, most pharmacogenetic tests are cost-effective or cost-saving [3,4,5,6]; nevertheless, in some therapeutic areas, further studies are required to determine cost-effectiveness [7].

The Clinical Pharmacology Department of Hospital Universitario de La Princesa (Madrid, Spain) is promoting an initiative for the implementation of pharmacogenetics: La Princesa University Hospital Multidisciplinary Initiative for the Implementation of Pharmacogenetics (PriME-PGx). Our initiative is not the first one to promote a similar action. Other initiatives, mainly from the United States, are active nowadays or finished recently. To our understanding, in Spain, the first and only implementation initiative is the MedeA initiative [8]. Briefly, this initiative intends to “integrate pharmacogenetics and other relevant information in a decision supporting tool to be used for individualized drug prescription during regular clinical practice within the context of e-health”. Our initiative is the second one of this kind in Spain and the third in Europe after the one promoted by the Ubiquitous Pharmacogenomics Consortium (U-PGx) [9]. The latter has not finished yet (ClinicalTrials.gov identifier NCT03093818). Nevertheless, the PriME-PGx initiative is novel and of great interest as it promotes the expansion of pharmacogenetics in the Hospital’s patients, in the general population and in the field of clinical trials; later in this text, it will be thoroughly described. However, to provide a context, in the following paragraphs some important implementation initiatives are described. van der Wouden et al., on behalf of the U-PGx, reviewed implementation projects and initiatives promoted over recent years [9], which are summarized as follows:

The Cleveland Clinic’s Personalized Medication Program: this program aims to implement a clinical decision support system (CDSS) to guide pharmacogenetics test ordering and provide gene-based dosing recommendations. This project is developed at Cleveland Clinic (USA) and focuses on *HLA-B**57:01/abacavir and *TPMT*/thiopurines drug-gene combinations.

The CLIPMERGE PGx initiative aims to clarify the requirements that will support the use of PGx in clinical care. This project is a pilot study (*n* = 1500) developed in Icahn School of Medicine at Mount Sinai, USA. It focuses on various Clinical Pharmacogenetics Implementation Consortium (CPIC) guidelines.

The eMERGE-PGx initiative intends to implement next generation sequencing (NGS) covering a variety of genes for patients who are likely to be prescribed a drug of interest in the following one to three years. It is conducted in many USA sites, including Boston Children’s Hospital and Mayo Clinic. A total of 84 genes are included covering various CPIC guidelines.

There are several other noteworthy initiatives: IGNITE (University of Florida, Indiana and Vanderbilt, USA), INGENIOUS (Indiana Institute of Personalized Medicine, USA), Personalized Medicine Program (University of Florida and Shands Hospital, USA), PG4KDS (St. Jude Children’s Research Hospital, USA), PGRN (University of Maryland, Florida, St Jude Children’s Hospital, among others), PREDICT (Vanderbilt University Medical Center, USA), RIGHT (Mayo Clinic, USA), The 1200 Patients Project (University of Chicago, USA), the Sanford Chip (Stanford Imagenetics Initiative, available at: genomes2people.org, accessed on 12 August 2021) [10], the implementation initiative of University of Colorado’s Biobank [11], among others. Furthermore, Borobia et al., from Hospital Universitario de La Paz, Madrid (Spain) published in 2018 their experience in clinical pharmacogenetics implementation [12]. Other projects have promoted the creation of other clinical decision support system (CDSS) tools to help with the integration of pharmacogenetic information in the clinical context, such as the FARMAPRICE CDSS and several others [13,14,15,16,17]. These were thoroughly revised by Hinderer et al. [18].

The constitution of the above-mentioned initiatives and our experience in clinical pharmacogenetics encouraged us to summarize our assistance activity since the constitution of our Pharmacogenetics Unit. Furthermore, we are currently promoting two novel sub-projects within the scope of the PriME-PGx initiative. As will be mentioned below, our project has some strengths and novelties compared to previous works. On the one hand, the historical achievements since the creation of our group are described, as well as the technological advances and milestones accomplished. On the other hand, the two mentioned starting ongoing sub-projects are presented: the PROFILE and the GENOTRIAL projects.

## 2. Historical Achievements

Founded in April 1857, the Hospital Universitario de La Princesa is a University Hospital of Madrid’s Health Service, Spain, that assists 323,000 people for basic specialties, and is the reference Hospital for nearly one million for highly complex specialties, such as neurosurgery, cardiac surgery, or thoracic surgery, among others. Annually, 16,000 hospital admissions are attended; 440,000 outpatients and 100,000 emergency patients are assisted. In 2018, the 2000th bone marrow transplant took place. The Clinical Pharmacology Department was established in 1995, thanks to the promotion of the Pharmacology department of Universidad Autónoma de Madrid. It offers the following healthcare services: therapeutic drug monitoring (TDM) (e.g., for antipsychotics and tyrosine kinase inhibitors, among others), general therapeutic consultations, Pain Management Unit consultations, evaluation of clinical study protocols, assistance in clinical trial design, promotion and performance, assistance in the evaluation of new drugs, medication errors, pharmacovigilance (i.e., adverse event reporting), and pharmacogenetics. Likewise, the Clinical Trials Unit of Hospital Universitario de La Princesa (UECHUP), part of the Clinical Pharmacology Department and of the PriME-PGx initiative, performs more than 20 clinical trials per year, ensuring a valuable source of data for pharmacogenetic research.

The Pharmacogenetics Unit was established in 2006 and has substantially evolved since then. Initially, the procedure for processing a pharmacogenetic test was as follows: a prescribing physician requested the genotype of a specific gene for the prescription of a specific drug, by means of a written paper form. This form was sent to our unit together with a blood sample from the patient. The sample was extracted, genotyped manually and, within 10–14 days, a result was provided. For its communication, a pharmacogenetic report was written, printed and hand-delivered to the prescriber.

The first pharmacogenetic test included in the Clinical Pharmacology Department test portfolio was *TPMT* genotyping, indicated for the prescription of thiopurines (i.e., azathioprine and 6-mercaptopurine). The first pharmacogenetic report was issued in June 2006. From 2006 to January 2021, 2434 patients were genotyped, approximately 203 patients per year. Among them, 1315 were genotyped for *HLA-B* (first genotyped in 2008)*,* for the screening of the abacavir hypersensitivity reaction (ABC-HSR), related to the *HLA-B**57:01 allele; 798 were genotyped for *IFNL3* rs12979860 and rs8099917 (first genotyped in 2011), for the prediction of response to pegylated interferon-α and ribavirin-based regimens for the treatment of hepatitis C [19]; 669 were genotyped for *TPMT* (first genotyped in 2006), and/or *NUDT15* (first genotyped in 2020), genes, for the prediction of thiopurine tolerability [20]; 187 were genotyped for *CYP2C19* (first genotyped in 2013), prior to clopidogrel prescription in patients undergoing a neurointerventional surgery, for the prevention of ischemic and haemorrhagic events [21]; 88 were genotyped for *DPYD* (first genotyped in 2013)*,* for the prediction of dihydropyrimidine (capecitabine or 5-fluoruracil) toxicity [22]; 171 patients in the Pain Management Unit or with other treatments (e.g., tamoxifen or antidepressants) received an individualised pharmacogenetic study which included CYP2D6 genotyping (first genotyped in 2015).

In the absence of recommendations from a Spanish society or consortium on pharmacogenetics, we initially adhered solely to CPIC pharmacogenetic guidelines [23,24]. With the progression of the discipline, other relevant societies emerged with clinical guidelines. Since 2017, for gene-drug pairs where CPIC has no guideline, the Dutch Pharmacogenetics Working Group (DPWG) recommendations [25] are applied. Should there be discrepancies between CPIC and DPWG recommendations for a particular drug-gene association, our consensus is to adhere to the CPIC recommendations. Furthermore, nowadays, some pharmacogenetic information is issued by regulatory agencies for certain drugs; in our case, AEMPS/EMA drug labels are fully addressed.

The portfolio of available tests changed over these years as we were able to overcome some of the above-mentioned barriers: our genotyping capacity was significantly improved and became more cost-effective: nowadays, we conduct array-based genotyping; the Spanish regulator (AEMPS) issued several genotyping recommendations (e.g., for siponimod and *CYP2C9* or for fluoropyrimidines and *DPYD*); and physicians and pharmacists are more aware of the usefulness of pharmacogenetics.

Table 1 shows all the genes included in our custom genotyping array (the Very Important Pharmacogene Open Array panel, VIPOA) with available clinical prescribing information and some of the important variants used to infer enzyme phenotype. Since CPIC provides comprehensive allele definition tables, functionality tables, etc., CPIC guidance, which is linked to PharmVAR, is followed in our pharmacogenetic unit. Consequently, all alleles considered “actionable” included in our array are described in CPIC/PharmVAR. Nevertheless, not all the variants in our array are clinically actionable (i.e., related to a pharmacogenetic phenotype that would require a modification of routine practice).

The above-mentioned progression brought our team to a position where we are able to promote the implementation of pharmacogenetics. To understand how, the main pharmacogenetic tests offered are described below along with their chronological implementation.

## 3. Relevant Pharmacogenetic Tests

### 3.1. TPMT, NUDT15 and Thiopurines

Thiopurine S-methyltransferase (TPMT) is an enzyme relevant to thiopurines metabolism (i.e., azathioprine and 6-mercaptopurine). Since 2004, it is known, with a high level of evidence, that its polymorphism affects its activity and that therapy should be individualized based on TPMT phenotype [26]. As mentioned earlier, *TPMT* was first genotyped in our pharmacogenetic unit in June 2006. *TPMT* *2, *3A, *3B, *3C, *3D, *4, *5, *6, *7 alleles were initially genotyped prospectively until 2010 and Sanger sequencing was outsourced. Since 2011, due to their low prevalence, *TPMT* *4, *5, *6, *7 were excluded from the analysis and Sanger sequencing was no longer performed. Manual qPCR was performed in a LightCycler thermal cycler (Roche Diagnostics, Barcelona, Spain) along with LightSNP probes designed by TIB Molbiol. Since 2020, *2, *3A, *3B, *3C and *4 alleles are included in a customized array which uses the TaqMan probes genotyping technology (QuantStudio 12k Flex, Open Array, ThermoFisher, Waltham, MA, USA).

In April, 2020, CPIC’s guideline update on *TPMT*-thiopurines was published, where recommendations based on *NUDT15* phenotype were included [20]. Since then, the *NUDT15**3 allele was included in our portfolio. The results of the 669 available genotypes since 2006 are shown in Table 2, being 92.4% of them normal metabolizers (NMs) (e.g., *1/*1), 7.0% intermediate metabolizers (IMs) (e.g., *1/*3A) 0.3% poor metabolizers (PMs) (e.g., *2/*2) and 0.3% indeterminate (e.g., *1/*8). Thanks to this pharmacogenetic test, 7.3% of potential thiopurine toxicities were avoided or reduced. All *NUDT15* genotypes available (*n* = 47) were *1/*1.

### 3.2. HLA-B

The human leukocyte antigen (HLA) is a combination of surface proteins located in every cell in the organism with a modulating function for the immune system. Several *HLA-A* and *HLA-B* alleles are related to drug toxicity. Probably, the best described association in the literature is the hypersensitivity reaction to abacavir (ABC-HSR) in carriers of the *HLA-B**57:01 allele. The CPIC guideline for this gene-drug pair was first published in 2012 [27]. In our pharmacogenetic unit, the first *HLA-B* genotyping test was performed in June 2008, as the evidence for this gene-drug pair was already high (i.e., the PREDICT-1 study had been published on February 2008) [28]. Initially, our genotyping strategy comprised HLA-B genotyping using the INNO-LiPA HLA-B Update Plus (Fujirebio, Tokyo, Japan) kit in a SimpliAmp thermal cycler (ThermoFisher, Waltham, MA, USA) and an automated reverse-hybridation AutoLipa system (Fujirebio, Tokyo, Japan). For *HLA-B**57 carriers, Sanger sequencing was outsourced for the confirmation of the HLA-B*57:01 sub-allele. Since 2008, until late 2019, a total of 1229 tests were performed. The vast majority of these tests had been requested by a physician willing to prescribe abacavir. However, some requests aimed at *HLA-B**58:01 (related to allopurinol toxicity) [29] or *HLA-B**15:02 (related to oxcarbazepine toxicity) [30]. The most prevalent *HLA-B* allele was B*44 (13.3%) (Table 3). A total of 63 B*57 alleles were observed, among which 50 were B*57:01 heterozygous, one was B*57:01 homozygous, 7 were B*57:03 heterozygous, 2 were B*57:02 heterozygous, 1 was B*57:16 heterozygous and one was B*57:07 heterozygous. *HLA-B**57:01 positives (heterozygous or homozygous) supposed 4.15% of the population. Since 2020, another 60 samples were genotyped for *HCP5* rs2395029, a surrogate marker for *HLA-B**57:01 screening, being 58 of them negative and 2 of them positive. Before implementing this SNP in our routine clinical practice, we validated the linkage disequilibrium in our population between this *locus* and *HLA-B**57:01; we confirmed the association to be complete [31]. Overall, *HLA-B**57:01 positives suppose 4.11% of the population receiving health care at our hospital. Thence, the ABC-HSR was avoided in >4% of patients receiving abacavir, which implies approximately half of the potential expected toxicities. Finally, *HLA-B**58 allele had a prevalence of 2.4% in our population, related to allopurinol-induced dermatological toxicity [32] and *HLA-B**15 allele of 6.6%, related to carbamazepine-associated cutaneous adverse reactions [30].

### 3.3. IFNL3 (IL28B)

In 2009, *IFNL3* genotype (*IL28B*) was found to be the best predictor of response to ribavirin (RBV) and pegylated interferon alpha (PEG-IFN-α) for the management of patients infected with hepatitis C virus Genotype 1 [33,34]. The first test in our pharmacogenetics unit for *IFNL3* rs12979860 and rs8099917 was performed in March 2011. Initially, LightSNP probes designed by TIB Molbiol (Madrid, Spain) were used for qPCR genotyping in a LightCycler instrument (Roche Diagnostics, Barcelona, Spain) and since 2020, these SNPs are included in our Open Array customized array. Genotyping of this variant was initially a requirement of the Spanish Ministry of Health for prescribing telaprevir and boceprevir in combination with pegylated interferon and ribavirin in patients with a low likelihood of achieving a sustained viral response. A total of 792 patients were genotyped and the results were as follows: 266 of them carried the *IFNL3* rs12979860 C/C genotype (33.6%), 409 the C/T genotype (51.6%) and 117 the T/T genotype (14.8%); 404 carried the rs8099917 T/T genotype (51.0%), 343 the G/T genotype (43.3%) and 45 the G/G genotype (5.7%). Currently, this test is rarely requested due to the disuse of these drugs in favor of direct antivirals.

### 3.4. CYP2C19

The cytochrome P450 isoform 2C19 (CYP2C19) metabolizes several relevant drugs like antidepressants, protein pump inhibitors and clopidogrel, among others [35]. The polymorphism of this gene is related to phenotypic variability in CYP2C19-mediated metabolism. The first test was performed in our pharmacogenetic unit in June 2013. In our hospital, this test is mainly performed for the prevention of atherothrombotic and thromboembolic events in patients with carotid, vertebral or cranial artery stent implantations [36,37]. Since 2013, a total of 188 patients were genotyped for *CYP2C19**2, *3, and *17 and since 2020, for *4, *5, *6, *7, *8 and *35, being the results as follows: 80 patients (42.6%) were NMs, 47 (25%) were rapid metabolizers (RM), 48 (25.5%) were IMs, 9 (4.8%) were ultrarapid metabolizers (UMs) and 4 (2.1%) were PMs (Table 4). Clopidogrel may not be used for IMs and PMs [21], therefore, for >27% of patients at risk for cardiovascular events, the drug was switched to prasugrel or ticagrelor. Additionally, our study associating the UM phenotype to bleeding risk [36] was well received by physicians at our hospital who, occasionally, also switched drugs for this phenotype.

### 3.5. DPYD

CPIC guideline on *DPYD* and fluoropirimidines (e.g., 5-fluoruracil or capceitabine) was first published in 2013 [38] and updated in 2017 [22]. The dihydropyrimidine dehydrogenase (DPD), encoded by the *DPYD* gene, significantly contributes to fluoropyrimidine metabolism. Decreased and no-function alleles are well described, causing the enzyme’s functional impairment and increasing the risk for severe toxicity. Dose adjustments or a change of the drug may be considered in patients with decreased or null enzyme activity. In November 2013, the first patient was genotyped for *DPYD* *2A, *13 and rs67376798. Since then and until 2020, only 16 patients were genotyped, the majority of them after severe toxicities. Since 2020, the following variants are genotyped: *2A, *7, *8, *10, *12, *13, rs67376798, rs115232898, rs6668296 and HapB3. Moreover, in May 2020, the Spanish Drug Agency (AEMPS) issued a recommendation for DPYD genotyping prior to fluoropyrimidine administration. Since then and until January 2021, 72 patients were genotyped for the indicated variants. Its implementation in clinical practice is expanding rapidly nowadays, as shown in Figure 1.

All of them were *1/*1 except for two subjects who were *1/*HapB3. In addition, since 2021, we perform Sanger sequencing of the entire gene for noncarriers of the genotyped variants who suffered severe toxicity. We are currently conducting a retrospective case-control study which will be published soon where we show novel *DPYD* variants related to capecitabine and 5-FU toxicity.

### 3.6. Pain Management Unit: Towards Complete Pharmacogenetic Reports

CPIC’s guideline on codeine and *CYP2D6* was published in 2012 [39]. Although this guideline had no specific therapeutic recommendations for tramadol based on CYP2D6 phenotype, several statements raised concerns about tramadol effectiveness in PMs and about tramadol toxicity in UMs. *CYP2D6* genotyping for patients at the Hospital’s Pain Management Unit started in July 2015. Initially, the following variants were genotyped: *3, *4, *5 (deletion), *6, *7 and *9. Since 2020, the following additional ones were genotyped: *10, *12, *14, *17, *19, *29, *41 *54 and *59. Table 2 shows the results for the 175 patients genotyped in this period. A total of 8 of them (4.6%) were UMs 102 were NMs (58.0%), 50 were IMs (28.7%) and 10 were PMs (5.7%). Finally, five individuals presented conflictive phenotypes. These patients all had three gene copies and showed allele heterozygosis. However, without a digital PCR, it was not possible to determine which of the alleles was duplicated. Three patients showed the (*1/*4)xN genotype, therefore, they could be NMs or IMs; one patient showed the (*1/*9)xN genotype and another showed the (*1/*41)xN, therefore, both could be UMs or NMs. For these five patients, a digital PCR should be performed in order to unequivocally infer CYP2D6 phenotype (Table 5).

Meanwhile, CPIC’s guideline on nonsteroidal anti-inflammatory drugs (NSAIDs) was published on March 2020 [40]. This meant that, for patients treated at the Pain Management Unit, two actionable pharmacogenetic tests were available from which they could benefit (*CYP2C9* and *CYP2D6* for NSAIDs and tramadol, respectively). This situation rendered obsolete the working procedure in which, for each patient, a specific pharmacogenetic test for a gene or drug was requested. Given our advances in genotyping technology and the greater pharmacogenetic knowledge available, lots of useful information were generated and not informed for the benefit of patients. Not only was important pharmacogenetic information related to their disease being generated, but a battery of pharmacogenes related to dozens of drugs and pathologies was also being genotyped. However, at this point, only individual gene-drug pairs were reported.

This motivated the establishment of the PriME-PGx initiative with two starting projects aimed at the expansion of clinical pharmacogenetics. The first one, the PROFILE project, in which specific pharmacogenetic profiles were created for specific therapeutic areas. Not only did this change the way pharmacogenetic results were reported, but also promoted the expansion of pharmacogenetic knowledge at our hospital. Briefly, instead of reporting individual gene-drug pairs, several of them were compiled in specific reports for each hospital department. The second one, the GENOTRIAL project, in which a report of clinically relevant pharmacogenetic findings is provided to any healthy volunteer consenting participation for pharmacogenetic research at the Clinical Trials Unit of Hospital Universitario de La Princesa (UECHUP). Both projects will be described in depth in the following sections.

## 4. The PROFILE Project

Actionable pharmacogenetic tests are nowadays directed to prescribers at the different Departments of our Hospital. Seven pharmacogenetic profiles were created based on the specific requirements of seven hospital departments. The full description of the seven profiles is shown in (Appendix A). Briefly, they are described as follows:Pain Management (PMU) profile: this profile includes evident drug-gene associations for anti-inflammatory and analgesic drugs (e.g., tramadol-*CYP2D6* and NSAIDs-*CYP2C9*) and other less evident pairs: antidepressants, statins or antiepileptic drugs (Appendix A).Oncology (ONC) profile: this profile includes evident drug-gene associations for antineoplastic drugs (e.g., DPYD and 5-fluorouracil, *CYP2D6* and tamoxifen or *UGT1A1* and irinotecan), immunosuppressants (e.g., *TPMT*/*NUDT15* for azathioprine and mercaptopurine and *CYP3A5* for tacrolimus) and other less evident pairs: tramadol, codeine, ondansetron or tropisetron (Appendix A).Neurology-psychiatry (NEU) profile: this profile includes evident drug-gene associations for antipsychotics (e.g., CYP2D6 and aripiprazole), selective serotonin reuptake inhibitors (SSRIs) (e.g., *CYP2D6* and fluvoxamine or *CYP2C19* and citalopram), tricyclic antidepressants (e.g., CYP2D6 and desipramine or CYP2C19 and imipramine), *CYP2C9*-siponimod and antiepileptic drugs (e.g., *HLA-B**15 and A*31 for carbamazepine) (Appendix A).Immunosuppressants (IMS) profile: this profile includes associations for immunosuppressants exclusively (e.g., *TPMT/NUDT15* for azathioprine and mercaptopurine and *CYP3A5* for tacrolimus) (Appendix A).Infectious Diseases (INF) profile: this profile includes evident drug-gene associations for anti-infectious agents (e.g., *HLA-B* for abacavir, *DPYD* for flucytosine, *IFNL3* for ribavirin or peg-α-2a/2b interferon, *UGT1A1* for atazanavir, *CYP2B6* for efavirenz and *CYP2C19* for voriconazole) (Appendix A).Gastroenterology (DIG) profile: this profile includes an evident drug-gene association, i.e., *CYP2C19* and protein pump inhibitors (PPIs) (e.g., omeprazole) and other less evident drug-gene pairs (*CYP2C19*-clopidogrel, *TPMT/NUDT15* for azathioprine and mercaptopurine or *CYP2C9*, *CYP4F2* and *VKORC1* for warfarin and acenocumarol) (Appendix A).Cardiovascular medicine (CAR) profile: this profile includes evident drug-gene associations for agents related to cardiovascular or blood function (e.g., *SLCO1B1* for statins or *CYP2C19* for clopidogrel and *CYP2C9*, *CYP4F2* and *VKORC1* for warfarin and acenocumarol) and other less evident drug-gene pairs (e.g., *CYP2C19*-PPIs) (Appendix A).

As previously mentioned, the way pharmacogenetic tests were requested and communicated changed significantly. With the PROFILE project, the methodology was modernized. No more applications for pharmacogenetic tests were processed on written paper. Nowadays, the clinical record allows requesting pharmacogenetic tests electronically. Physicians can select pharmacogenetic profiles which contain the test they would like to request along with several other tests that are related to their medical specialty. Array genotyping allows designing a panel of relevant pharmacogenes which covers all of the pharmacogenetic profiles mentioned. Hence, information for all of these genes is obtained and can be used for the benefit of the patient. In addition, an alert system for relevant pharmacogenetic findings was implemented in patient’s medical record, regardless the requested profile. For instance, should a physician request the CAR profile, a report in pdf format will be uploaded to their medical record with all relevant detailed pharmacogenetic information. Further, if any other relevant information is discovered, (e.g., the patient is a *DPYD* PM or carries the *HLA-B**57:01 allele), simplified alerts will be added to the patient’s medical record, similar to an allergy alert (e.g., “*DPYD PM* alert: prescription of capecitabine and 5-fluorouracil”; *HLA-B*57:01* alert: do not prescribe abacavir).

It is worth noting that, since 2020, the Clinical Pharmacology Department offers a Pharmacogenetics consultation every Friday. This is aimed at polymedicated patients who require an individualized analysis of interactions and pharmacogenetic findings to minimize the toxicity of their pharmacotherapy and increase its effectiveness.

### 4.1. The GENOTRIAL Project

Our Pharmacogenetics Unit works in tight collaboration with UECHUP, as both groups are part of the Clinical Pharmacology Department of the Hospital. Dozens of bioequivalence and other Phase 1, 2 and 3 clinical trials are conducted in this unit every year. Thanks to the availability of these data (i.e., the outcomes of the clinical trials) and the informed consent of the healthy volunteers, a valuable source of information is available for pharmacogenetic research. In the last decade, our group contributed to the expansion of available pharmacogenetic knowledge with numerous publications [36,37,41,42,43,44,45,46,47].

Since the modernization of our genotyping technology, something similar happened with healthy volunteers to what is described earlier for patients. Their genotyping for research purposes was yielding clinically relevant information for them and they were not being informed about it. From an ethical perspective, we believed that associations relevant to the future pharmacotherapy of patients should be reported. This encouraged the creation of the GENOTRIAL project, which aimed at providing a pharmacogenetic report of relevant findings to any healthy volunteer who desired it. Firstly, the most obvious is the positive impact this activity has on healthy volunteers’ health, especially in the long term. Knowing how they will respond to drugs will help in the future to assign personalized therapy to them. Secondly, from the pharmaceutical industry’s point of view, this is of great benefit: knowing groups of healthy volunteers with homogeneous metabolism could help to reduce sample sizes in bioequivalence trials [48]. Not only is this economically beneficial, but it is also an ethical imperative. In our clinical trials unit, clinical trials of this type are already ongoing, e.g., in one clinical trial, only CYP2D6 PMs are enrolled.

### 4.2. Automation

Notwithstanding, the generation of such large numbers of reports, both the new profiles from the PROFILE project and the healthy volunteers from the GENOTRIAL project, posed an obvious problem: no time or personnel enough was available for the manual preparation of such reports. Consequently, in collaboration with the Hospital’s Bioinformatics Unit, we promoted the automation of report generation through an R script. Figure 2 shows the workflow at our laboratory, which allows the fast integration of all pharmacogenetic data from patients and healthy volunteers, the automatic drafting of the complete pharmacogenetic report, profile reports and the relevant findings report. Appendix A show an example of a complete pharmacogenetic report and a relevant finding report.

A study is currently underway on the expectations of patients and healthy volunteers in our hospital to be informed of their pharmacogenetic reports. It will, therefore, be a cross-sectional study to PROFILE and GENOTRIAL. It is expected to be published before the end of 2021.

## 5. Conclusions and Future Perspective

This document presents the assistance experience of a multidisciplinary group specialized in clinical pharmacogenetics from 2006 to 2020. Not only did we implement pharmacogenetic testing in our hospital, but we are actively participating in its implementation at regional and national level. This justifies the creation of the PriME-PGx initiative, a pioneer project in our country and in Europe. Our initiative initially promotes the PROFILE and GENOTRIAL projects, which will contribute in the short term to the expansion of pharmacogenetic knowledge among professionals, the general population and throughout the field of clinical trials.

## Figures and Tables

**Figure 1 jcm-10-03772-f001:**
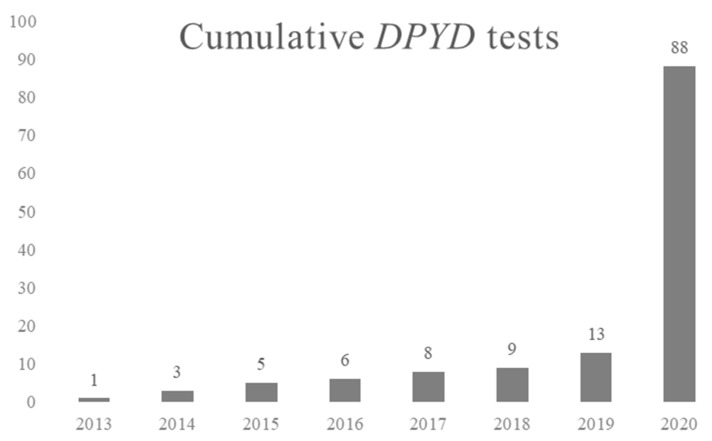
Cumulative *DPYD* genotyping tests performed at the Pharmacogenetics Unit, Hospital Universitario de La Princesa.

**Figure 2 jcm-10-03772-f002:**
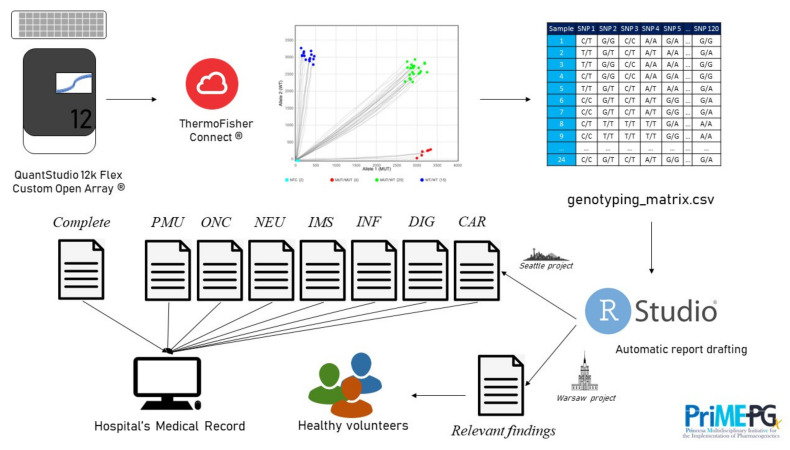
Laboratory workflow for the automatic generation of the different pharmacogenetic reports involved in the PROFILE and GENOTRIAL projects.

**Table 1 jcm-10-03772-t001:** Genes and variants included in the VIPOA genotyping panel.

Gene	Allele	SNP (rs)	Ancestral	Mutant	Defines Actionable ^#1^ Allele?
*CYP4F2*	Not defined	rs2108622	C	T	YES
*CYP2B6*	Multiple	rs3745274	G	T	YES
	Multiple	rs3211371	C	T	YES
	Not defined	rs4803419	C	T	NO
	Multiple	rs2279343	A	G	YES
	*22	rs34223104	C	T	YES
	*18, *16	rs28399499	T	C	YES
*CYP2C9*	*2	rs1799853	C	T	YES
	*3	rs1057910	A	C	YES
	*5	rs28371686	C	G	YES
	*8	rs9332094	T	C	YES
	*8	rs7900194	T	G	YES
	*11	rs28371685	C	T	YES
*CYP2C19*	*2	rs4244285	G	A	YES
	*3	rs4986893	G	A	YES
	*4	rs28399504	A	G	YES
	*6	rs72552267	G	A	YES
	*5	rs56337013	C	T	YES
	*7	rs72558186	T	C	YES
	*8	rs41291556	T	C	YES
	*9	rs17884712	G	A	YES
	*17	rs12248560	C	T	YES
	*2, *35	rs12769205	A	G	YES
*CYP2D6*	*3	rs35742686	T	-	YES
	*4	rs3892097	C	T	YES
	*6	rs5030655	A	-	YES
	*7	rs5030867	T	G	YES
	*8	rs5030865	C	A	YES
	*9	rs5030656	CTT	-	YES
	*10, *4	rs1065852	C	T	YES
	*10	rs1135840	C	G	YES
	*12	rs5030862	C	T	YES
	*14	rs5030865	C	T	YES
	*15	rs774671100	A	-	YES
	*17	rs28371706	G	A	YES
	*19	rs72549353	AGTT	-	YES
	*29	rs59421388	G	A	YES
	*41	rs28371725	C	T	YES
	*56B	rs72549347	G	A	YES
	*59	rs79292917	C	T	YES
*CYP3A5*	*3	rs776746	T	C	YES
	*6	rs10264272	C	T	YES
	*7	rs41303343	A	-	YES
*DPYD*	*2A	rs3918290	C	G/T	YES
	*12	rs1057519962	G	A	YES
	*12	rs1057519962	G	T	YES
	*10	rs1801268	C	A	YES
	*7	rs72549309	ATGAATGA	ATGA	YES
	*8	rs1801266	G	A	YES
	*13	rs55886062	A	C/T	YES
	HapB3	rs75017182	G	C	YES
	HapB3	rs75017182	G	T	YES
	c.2846A > T	rs67376798	T	A	YES
	c.557A > G	rs115232898	T	C	YES
	HapB3 (tag)	rs56038477	C	T	YES
	c.680 + 139G > A	rs6668296	T	C	NO
*HCP5*	*HLA-B**57:01	rs2395029	T	G	YES ^#2^
*HCP5*	*HLA-B**57:01	rs2395029	T	G	YES
*IL28B*		rs12979860	T	C	YES
*TPMT*	*2	rs1800462	C	G	YES
	*3B, *3A	rs1800460	G	A	YES
	*3C, *3A	rs1142345	A	G	YES
	*4	rs1800584	C	T	YES
	*11	rs72552738	C	T	YES
*REP TPMT*	*2	rs1800462	C	G	YES
	*3B, *3A	rs1800460	G	A	YES
*NUDT15*	*3	rs116855232	C	T	YES
*VKORC1*	(−1639G > A)	rs9923231	C	T	YES
		rs9934438	G	A	NO
		rs7294	C	T	NO
*UGT1A1*	*6	rs4148323	G	A	YES
	*80	rs887829	C	T	YES ^#3^
*HLA-A3101*		rs1061235	A	T	YES ^#4^
*SLCO1B1*	*5	rs4149056	T	C	YES
	*1b	rs2306283	G	A	YES
	c.−910G > A	rs4149015	G	A	YES
	*2	rs56101265	T	C	YES
	*3	rs56061388	T	C	YES
	*6	rs55901008	T	C	YES
	*9	rs59502379	G	C	YES
	*10	rs56199088	A	G	YES
		rs11045879	T	C	NO
*CYP1A2*	*1C	rs2069514	G	A	NO
	*1F	rs762551	A	C	NO
	*1B	rs2470890	T	C	NO
*CYP2A6*	*9	rs28399433	A	C	NO
*CYP2C8*	*2	rs11572103	T	A	NO
	*3	rs10509681	T	C	NO
		rs11572080	C	T	NO
	*4	rs1058930	G	C	NO
*CYP3A4*	*3	rs4986910	A	G	NO
	*2	rs55785340	A	G	NO
	*6	rs4646438	T	TT	NO
	*18	rs28371759			NO
	*22	rs35599367	C	T	NO
*ABCB1*	C3435T	rs1045642	C	T	NO
	G2677 T/A	rs2032582	C	A	NO
	G2677 T/A	rs2032582	C	T	NO
	C1236T	rs1128503	G	A	NO
*TBL1Y (SEX)*		rs768983			NO
*ABCG2*		rs2231142	G	T	NO
*ABCC2*		rs2273697	G	A	NO
*COMT*		rs4680	G	A	NO
		rs13306278	C	T	NO
*OPRM1*		rs1799971	A	G	NO
*SLC22A1*	*2	rs72552763	GAT	-	NO
	*3	rs12208357	C	T	NO
	*5	rs34059508	G	A	NO
*UGT2B15*		rs1902023	A	C	NO
*RARG*		rs2229774	G	A	NO
*SCL28A3*		rs7853758	G	A	NO
*UGT1A4*		rs2011425	T	A	NO
*UGT1A4*		rs2011425	T	G	NO
*EPHX1*		rs2234922	A	G	NO
		rs1051740	T	C	NO
*MTHFR*		rs1801133	G	A	NO
*XPC*		rs2228001	T	G	NO
*ERCC1*		rs11615	A	G	NO
*ERCC1*		rs3212986	A	C	NO
*XRCC1*		rs25487	C	T	NO

#^1^: The term “actionable allele” refers to variants related to, or defining alleles related to phenotypes potentially associated with a clinical recommendation issued by the Clinical Pharmacogenetics Implementation Consortium (CPIC) or the Dutch Pharmacogenetics Working Group (DPWG). ^#2^: The linkage disequilibrium (LD) between this variant and *HLA-B**57:01 has been validated and may be used as a surrogate biomarker. ^#3^: According to CPIC’s guideline on UGT1A1 and irinotecan, *UGT1A1**80 is in very high LD with *28 and can be considered a surrogate marker. ^#4^: This variant has been proposed as a surrogate biomarker for *HLA-A**31:01 but requires LD validation.

**Table 2 jcm-10-03772-t002:** Prevalence of *TPMT* genotypes in a Spanish population.

Genotype	Count	%	Phenotype
*1/*1	618	92.4	NM
*1/*3A	33	4.9	IM
*1/*2	9	1.3	IM
*1/*3C	5	0.8	IM
*2/*2	1	0.2	PM
*3A/*3C	1	0.2	PM
*1/*8	1	0.2	Ind
*1/*19	1	0.2	Ind
Total	669	100	

NM: normal metabolizer; IM: intermediate metabolizer; PM: poor metabolizer; Ind: indeterminate.

**Table 3 jcm-10-03772-t003:** Prevalence of *HLA-B* alleles in a Spanish population.

*HLA-B* Allele	Count	%	*HLA-B* Allele	Count	%
*44	327	13.3	*13	42	1.7
*35	274	11.1	*41	30	1.2
*7	187	7.6	*45	29	1.2
*51	187	7.6	*55	27	1.1
*18	173	7.0	*37	24	1.0
*15	162	6.6	*48	16	0.7
*14	136	5.5	*42	12	0.5
*40	128	5.2	*47	10	0.4
*8	105	4.3	*81	8	0.3
*49	81	3.3	*78	5	0.2
*39	78	3.2	*56	4	0.2
*27	69	2.8	*73	2	0.1
*57	63	2.6	*95	2	0.1
*53	62	2.5	*4	1	0.0
*58	57	2.3	*46	1	0.0
*38	55	2.2	*54	1	0.0
*50	52	2.1	*67	1	0.0
*52	47	1.9	Total *	2458	100%

* Total: refers to total number of alleles (2*n*); total number of patients (*n*) was 1229.

**Table 4 jcm-10-03772-t004:** Prevalence of *CYP2C19* genotypes in a Spanish population.

Genotype	Count	%	Phenotype
*1/*1	80	42.6	NM
*1/*17	47	25.0	RM
*1/*2	39	20.7	IM
*2/*17	9	4.8	IM
*17/17	9	4.8	UM
*2/*2	4	2.1	PM
Total	188	100	

UM: ultrarapid metabolizer; RM: rapid metabolizer; NM: normal metabolizer; IM: intermediate metabolizer; PM: poor metabolizer

**Table 5 jcm-10-03772-t005:** Prevalence of *CYP2D6* genotypes in a Spanish population.

Genotype	CNV	Count	%	Phenotype
*1/*5	1 copy (6.3%)	9	5.1	IM
*4/*5		2	1.1	PM
*1/*10	2 copies (85.7%)	82	46.9	NM
*1/*4		26	14.9	IM
*1/*9		10	5.7	NM
*1/*41		6	3.4	NM
*1/*6		6	3.4	IM
*4/*4		5	2.9	PM
*1/*17		3	1.7	NM
*4/*9		3	1.7	IM
*10/*10		1	<1%	IM
*3/*3		1	<1%	PM
*3/*4		1	<1%	PM
*4/*10		1	<1%	IM
*4/*41		1	<1%	IM
*4/*6		1	<1%	PM
*9/*9		1	<1%	IM
*4/*17		1	<1%	IM
*41/*41		1	<1%	IM
(*1/*1) xN	3 copies (8.0%)	8	4.6	UM
(*1/*4) xN		3	1.7	NM or IM
(*1/*9) xN		1	<1%	NM or UM
(*1/*10) xN		1	<1%	NM
(*1/*41) xN		1	<1%	NM or UM
Total		175	100%	

UM: ultrarapid metabolzier; RM: rapid metabolizer; NM: normal metabolizer; IM: intermediate metabolizer; PM: poor metabolizer. CNV: number of gene copies. xN: number of copies is 3 or greater but the duplicated allele is un-known.

## Data Availability

All data are published in the main text and Appendix A.

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
