# Peer review of "PriME-PGx: La Princesa University Hospital Multidisciplinary Initiative for the Implementation of Pharmacogenetics"

_jcm, 2021, doi:10.3390/jcm10173772_

Round 1
Reviewer 1 Report
Zubiaur et al describes La Princesa University Hospital Multidisciplinary Initiative for the Implementation of Pharmacogenetics (PriME-PGx) and present objectives of two affiliated projects: PROFILE project and GENOTRIAL project. The manuscript is well written although a few issues need to be addressed.
Major revisions:
- Are missing important references on pharmacogenetic CDSS (clinical decision support system) and recent European experiences with them. Please implement.
- Important references are also missing regarding the pharmacoeconomic repercussion of a fully PGx implementation in the clinical practice. Please add references from European cost-evaluation of Pharmacogenetics in clinical practice.
- Line 44. Which kind of societies? Scientific societies? Every PGx guidelines present gene-drug pairs with high level of evidence? Please discuss, it is more articulated than described.
- Line 45. “The cost of the necessary equipment… was significantly reduced”. When? Please articulate. Line 93-94 not clear, please rephrase.
- Table 1. need to show the complete panel SNP. You choose whether to show allele or SNP rs. It is not clear to me whether UGT1A1*28 is included in the panel or not. Could you please also insert another column specifying the presence of such SNP in CPIC or DPWG guidelines?
- Lines 258-259. Why DPYD*7, *8, 10, *12 *13, rs115232898, rs6668296 have been included? Are they cited in CPIC or DPWG guidelines? See point above. It is most needed to specify where the panel adopted and described has been designed.
- Line 287. Could you please explain how digital PCR could unequivocally indicate CYP2D6 genotype? It is not clear to me.
- Line 322. Why is not included tamoxifen in the oncology profile? CPIC guidelines are present too.
Minor revisions:
- Line 18. Europe includes Spain. Please correct.
- Many typos are present. A few are following:
- Genotype line 284
- Stable 1. DPYD > 5-fluorouracil;
- Stable 1. CYP2C9 > ibuprofen;
- Stable 1. CYP2C9+HLAB > Fenitoin;
- Stable 1. Infectious disease;
- Stable 1. DPYD > flucytosine
Author Response
Dear reviewer,
You will find a letter attached where you will find a response to all your comments and those of the other reviewers.
Thank you very much for your positive contribution to this manuscript.
Regards,
Pablo Zubiaur.

Reviewer 2 Report
Nice description of laboratory technologies used and sample results in Spanish population. Reviewer commends authors for their implementation work and improvement and growth of program throughout the years. A few suggestions to improve the clarity of the work:
Line 80: Currently states Boston Children’s Hospital or Mayo Clinic. Should “or” be replaced by “and”
Line 82: You state “INGNITE” and I believe this should be “IGNITE”
When listing noteworthy initiatives, you may want to check out University of Pittsburgh; Sanford Health Imagenetics and the Sanford Chip (PMID: 31453774) (Precision Population Medicine in Primary Care: The Sanford Chip Experience (genomes2people.org)); and University of Colorado (PMID: 32077359)
Line 131: 2 commas present after in 2011)
Paragraph starting at Line 141: what is the process if there are conflicting recommendations from CPIC/DPWG/regulatory agencies?
LINE 235: May just be reviewer preference but consider “and” in place of “or”
Line 243: not sure if contraindicated is the best word here unless included in package labeling?
In general, ”Moreover” is used multiple times in the manuscript. Is there a way the authors can find another word or organize paragraph/sentence structure to not need to use the word so many times?
Author Response

(The authors gave the same response as above.)

Round 2
Reviewer 1 Report
Another important reference on European pharmacogenetic CDSS is missing (PMID: 30987397). The same for pharmacoeconomic studies and their repercussion of PGx implementation in the clinical practice (PMID: 31155283; PMID: 30339275).
Author Response
Thank you for your suggestions; we have included the proposed references in the manuscript as indicated in the attached letter.
